# What Causes Polysemanticity? An Alternative Origin Story of Mixed Selectivity from Incidental Causes

## Abstract

Polysemantic neurons – neurons that activate for a set of unrelated features – have been seen as a significant obstacle towards interpretability of task-optimized deep networks, with implications for AI safety. The classic origin story of polysemanticity is that the data contains more "features" than neurons, such that learning to perform a task forces the network to co-allocate multiple unrelated features to the same neuron, endangering our ability to understand networks' internal processing. In this work, we present a second and non-mutually exclusive origin story of polysemanticity. We show that polysemanticity can arise incidentally, even when there are ample neurons to represent all features in the data, a phenomenon we term *incidental polysemanticity*. Using a combination of theory and experiments, we show that incidental polysemanticity can arise due to multiple reasons including regularization and neural noise; this incidental polysemanticity occurs because random initialization can, by chance alone, initially assign multiple features to the same neuron, and the training dynamics then strengthen such overlap. Our paper concludes by calling for further research quantifying the performance-polysemanticity tradeoff in task-optimized deep neural networks to better understand to what extent polysemanticity is avoidable.

## 1 Introduction

Deep neural networks are widely regarded as difficult to mechanistically understand, especially at the massive scales of modern frontier models. Such lack of interpretability is increasingly viewed as a serious concern in AI Safety since highly capable models might behave in unpredictable and undesirable ways Hendrycks et al. (2023); Ngo et al. (2022). One outstanding challenge preventing better mechanistic interpretability of networks is *polysemanticity*, a phenomenon whereby individual neurons activate for unrelated input "features" Olah et al. (2017; 2020). This phenomenon, why it occurs and how to interpret networks' computation nonetheless has also been studied for decades by neuroscientists under the term of "mixed selectivity", e.g., Asaad et al. (1998); Mansouri et al. (2006); Warden & Miller (2007); Rigotti et al. (2013); Barak et al. (2013); Raposo et al. (2014); Fusi et al. (2016); Parthasarathy et al. (2017); Lindsay et al. (2017); Zhang et al. (2017); Johnston et al. (2020).

A leading hypothesis for why neural networks learn polysemanticitic representations is out of necessity: if a task contains many more features than neurons, then achieving high performance at the task might force the network to co-allocate unrelated features to the same neuron Elhage et al. (2022). While intuitive and persuasive, in this work, we propose a second and non-mutually exclusive hypothesis: that polysemanticity might be caused by non-task factors in the training process. Because such factors are not necessary to perform the task well, we call this form *incidental polysemanticity*.

### 1.1 An alternative origin story

In this paper, we study two non-task factors that could produce incidentally polysemantic representations: $l_1$ regularization and neural noise. The intuition for why these factors would have such an effect is as follows: The reason neural networks can learn anything starting with completely random

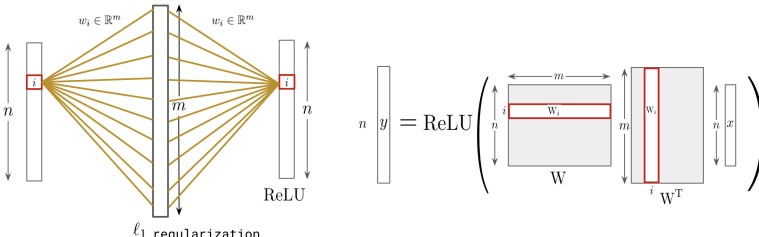

Figure 1: A visualization of the non-linear autoencoder setup with tied weights $W \in \mathbb{R}^{n \times m}$, a single hidden layer of size $m$, $\ell_1$ regularization with parameter $\lambda$, and a $\mathrm{ReLU}$ on the output layer.

weights is that, purely by random chance, some neurons will happen to be very slightly correlated[1] with some useful feature, and this correlation gets amplified by gradient descent until the feature is accurately represented. If, in addition to this, there is some incentive for activations to be sparse, then the feature will tend to be represented by a single neuron as opposed to a linear combination of neurons: this is a winner-take-all dynamic Oster et al. (2009).[2] When a winner-take-all dynamic is present, then by default, the neuron that is initially most correlated with the feature will be the neuron that wins out and represents the feature when training completes.

How often should we expect this incidental polysemanticity to happen? Suppose that we have $n$ useful features to represent and $m \geq n$ neurons to represent them with (so that it is technically possible for each feature to be represented by a different neuron). By symmetry, the probability that the $i^{\text{th}}$ and $j^{\text{th}}$ feature "collide", in the sense of being initially most correlated with the same neuron, is exactly $1/m$. And there are $\binom{n}{2} = n(n-1)/2$ pairs of features, so on average we should expect $\binom{n}{2} \times \frac{1}{m} = \frac{n(n-1)}{2m} = \Theta\left(\frac{n^2}{m}\right)$ collisions[3] overall. In particular, this means that (a) if $m \leq O(n)$ (i.e. the number of neurons is at most a constant factor bigger than the number of features), then $\Omega(n^2/n) = \Omega(n)$ collisions will occur: a constant fraction of all neurons will be polysemantic; (b) as long as $m$ is significantly smaller than $n^2$, we should expect several collisions to occur.

Our experiments with small autoencoders show that this is precisely what happens, and a constant fraction of these collisions do result in polysemantic neurons, despite the fact that there are enough neurons to avoid polysemanticity entirely. In the rest of this paper, we describe two simple models which exhibit incidental polysemanticity: one based on $l_1$ regularization (Section 2) and the other based on neural noise (Section 3). We study their sparsity and winner-take-all dynamics in mathematical detail, explore what happens over training when features collide, and confirm experimentally that the number of polysemantic neurons that are produced is a precise asymptotic match. In Appendix D, we show that even though these two cases are very different mathematically and even display different polysemantic configurations, their overall behavior is similar qualitatively. Finally, in Section 4 discuss implications for mechanistic interpretability and suggest interesting future work.

## 2 Incidental polysemanticity from regularization

As a first step, we show how polysemanticity can arise from a push for sparsity that is induced by $l_1$ regularization term on the representations.

---

[1] When we say a neuron is correlated with a feature, we formally mean that the neuron's activation is correlated with whether the feature is present in the input (where the correlation is taken over the data points).

[2] Analogous phenomena are known under other names, such as "privileged basis".

[3] Here, we define a "collision" as the event that two features $i$ and $j$ collide. So for example there is a three-way collision between $i$, $j$ and $k$, that would count as three collisions between $i$ and $j$, $i$ and $k$, and $j$ and $k$.

## 2.1 NETWORK AND DATA

We consider a model similar to the one in Elhage et al. (2022). It is a shallow nonlinear autoencoder with $n$ features (inputs or outputs), a weight tying between the encoder and the decoder (let $W \in \mathbb{R}^{n \times m}$ be those weights), uses a single hidden layer of size $m$ with $l_1$ regularization of parameter $\lambda$ on the activations, has a ReLU on the output layer with no biases anywhere, and is trained with the $n$ standard basis vectors as data (so that the "features" are just individual input coordinates): that is, the input/output data pairs are $(e_i, e_i)$ for $i \in [n]$, where $e_i \in \mathbb{R}^n$ is the $i^{\text{th}}$ basis vector. The shallow nonlinear autoencoder's output is computed as $y := \text{ReLU}(WW^\mathsf{T} x)$.

The main difference compared to the shallow nonlinear autoencoder from Elhage et al. (2022) is the addition of $l_1$ regularization. The role of the $l_1$ regularization is to push for sparsity in the activations and therefore induce a winner-take-all dynamic. We picked this model because it makes incidental polysemanticity particularly easy to demonstrate and study, but we do think the story it tells is representative (see Section 4 for more on this); for instance, even if $l_1$ regularization is not widely used in practice, recent work has also shown that other factors such as noisy data can implicitly induce sparsity-favoring regularization Bricken et al. (2023). We make the following assumptions on parameter values:

- the weights $W_{ik}$ are initialized to i.i.d. normals of mean $0$ and standard deviation $\Theta(1/\sqrt{m})$—so that the encodings $W_i \in \mathbb{R}^m$ start out with constant length.
- $m \geq n$ to make it clear that polysemanticity is not necessary in this setting.
- $\lambda \leq 1/\sqrt{m}$ so that the $l_1$ regularization doesn't kill all weights immediately.

## 2.2 POSSIBLE SOLUTIONS

Let $W_i \in \mathbb{R}^m$ be the $i^{\text{th}}$ row of $W$. It tells us how the $i^{\text{th}}$ feature is encoded in the hidden layer. When the input is $e_i$, the output of the model can then be written as

$$(\text{ReLU}(W_1 \cdot W_i), \dots, \text{ReLU}(W_n \cdot W_i)),$$

For this to be equal to $e_i$ we need $\|W_i\|^2 = 1^4$ and $W_i \cdot W_j \leq 0$ for $j \neq i$. Letting $f_k \in \mathbb{R}^m$ denote the $k^{\text{th}}$ basis vector in $\mathbb{R}^m$. There are both monosemantic and polysemantic solutions that satisfy these conditions:

- One solution is to simply let $W_i := f_i$: the $i^{\text{th}}$ hidden neuron represents the $i^{\text{th}}$ feature, and there is no polysemanticity.
- But we could also have solutions where two features share the same neuron, with opposite signs. For example, for each $i \in [n/2]$, we could let $W_{2i-1} := f_i$ and $W_{2i} := -f_i$. This satisfies the conditions because $W_{2i-1} \cdot W_{2i} = f_i \cdot (-f_i) = -1 \leq 0$.
- In general, we can have a mixture of these where each neuron represents either 0, 1 or 2 features, in an arbitrary order.

## 2.3 LEARNING DYNAMICS AND LOSS

Let us consider total squared error loss $\mathcal{L}$, which can be written as

$$\sum_i \left( \left(1 - \|W_i\|^2\right)^2 + \sum_{j \neq i} \text{ReLU}(W_i \cdot W_j)^2 + \lambda \|W_i\|_1 \right).$$

The training dynamics are

$$\frac{\mathrm{d}W_i}{\mathrm{d}t} := -\frac{\partial \mathcal{L}}{\partial W_i} = \underbrace{4(1 - \|W_i\|^2)W_i}_{\text{feature benefit}} - \underbrace{4 \sum_{j \neq i} \text{ReLU}(W_i \cdot W_j)W_j}_{\text{interference}} - \underbrace{\lambda \, \text{sign}(W_i)}_{\text{regularization}}$$

---

[4]We use $\|\cdot\|$ to denote Euclidean length ($l_2$ norm), and $\|\cdot\|_1$ to denote Manhattan length ($l_1$ norm).

where $t$ is the training time (which corresponds to the learning rate multiplied by the number of training steps). For simplicity, we'll ignore the constants 4 going forward[5]. It can be decomposed into three intuitive "forces" acting on the encodings $W_i$: (1)"feature benefit": encodings want to have unit length; (2) "interference": different encodings avoid pointing in similar directions; (3) "regularization": encodings want to have small $l_1$-norm (which pushes all nonzero weights towards zero with equal strength).

## 2.4 THE WINNING NEURON TAKES IT ALL

**Sparsity force**    For a moment, let's ignore the interference force, and figure out how (and how fast) regularization will push towards sparsity in some encoding $W_i$. Since we're only looking at feature benefit and regularization, the other encodings $W_j$ have no influence at all on what happens in $W_i$. Assuming $\|W_i\| < 1$, each weight $W_{ik}$ is pushed up with strength $\left(1 - \|W_i\|^2\right) W_{ik}$ by the feature benefit force and pushed down with strength $\lambda \operatorname{sign}(W_{ik})$ by the regularization.

Crucially, the upwards push is *relative* to how large $W_{ik}$ is, while the downwards push is *absolute*. This means that weights whose absolute value is above some threshold $\theta$ will grow, while those below the threshold will shrink, creating a "rich get richer and poor get poorer" dynamic that will push for sparsity. This threshold is determined by

$$(1 - \|W_i\|^2)W_{ik} = \lambda \operatorname{sign}(W_i) \iff |W_{ik}| = \frac{\lambda}{1 - \|W_i\|^2}$$

so letting $\theta := \frac{\lambda}{1 - \|W_i\|^2}$, we have

$$\frac{\mathrm{d}|W_{ik}|}{\mathrm{d}t} = \underbrace{(1 - \|W_i\|^2)|W_{ik}|}_{\text{feature benefit}} - \underbrace{\lambda \mathbf{1}[W_{ik} \neq 0]}_{\text{regularization}}$$

$$= \begin{cases} \underbrace{(1 - \|W_i\|^2)}_{\text{constant in } k} \underbrace{(|W_{ik}| - \theta)}_{\text{distance from threshold}} & \text{if } W_{ik} \neq 0 \\ 0 & \text{otherwise.} \end{cases}$$

We call this combination of feature benefit and regularization force the *sparsity* force. It uniformly stretches the gaps between (the absolute values of) different nonzero weights. Note that the threshold $\theta$ is not fixed: we will see that as $W_i$ gets sparser, $\|W_i\|^2$ will get closer to 1, which increases the threshold and allows it to get rid of larger entries, until only one is left. But how fast will this go?

**How fast does it sparsify?**    In order to track how fast $W_i$ sparsifies, we will look at its $l_1$ norm $\|W_i\|_1 = \sum_k |W_{ik}|$ as a proxy for how many nonzero coordinates are left. Indeed, we will have $\|W_i\| \approx 1$ throughout, so if $W_i$ has $m'$ nonzero values at any point in time, their typical value will be $\pm 1/\sqrt{m'}$, which means $\|W_i\|_1 \approx m'\frac{1}{\sqrt{m'}} = \sqrt{m'}$.

Since the sparsity force is proportional to $1 - \|W_i\|^2$, we need to get a sense of what values $\|W_i\|$ will take over time. As it turns out, $\|W_i\|$ changes relatively slowly, so we can get useful information by assuming the derivative $\frac{\mathrm{d}\|W_i\|^2}{\mathrm{d}t}$ is 0:

$$0 \approx \frac{\mathrm{d}\|W_i\|^2}{\mathrm{d}t} = 2\frac{\mathrm{d}W_i}{\mathrm{d}t} \cdot W_i = 2 \left( \underbrace{\left(1 - \|W_i\|^2\right) \|W_i\|^2}_{\text{from feature benefit}} - \underbrace{\lambda\|W_i\|_1}_{\text{from regularization}} \right),$$

which means $1 - \|W_i\|^2 \approx \frac{\lambda\|W_i\|_1}{\|W_i\|^2}$. Plugging this back into $\frac{\mathrm{d}\|W_i\|_1}{\mathrm{d}t} = \sum_k \frac{\mathrm{d}|W_{ik}|}{\mathrm{d}t}$ and using reasonable assumptions about the initial distribution of $W_i$, we can prove (see Appendix B for details)

---

[5]It's equivalent to making $\lambda$ four times larger and making training time four times slower.

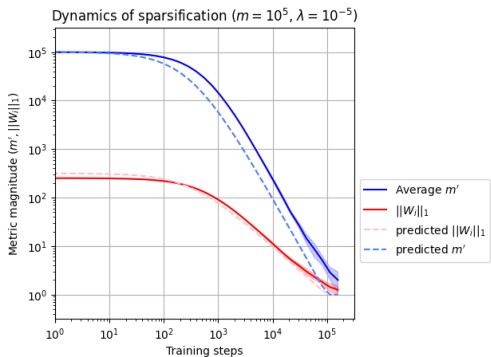

Figure 2: Number of non-zero coordinates $m'$ in $W_i$ and the value of $||W_i||_1$ plotted with training steps. The simulation confirms the speed of sparsification hypothesis.

that $\|W_i\|_1$ will decrease proportionally to $1/\lambda t$ with training time $t$:

$$\|W_i(t)\|_1 = \begin{cases} \Theta(\sqrt{m}) & t \le \frac{1}{\lambda\sqrt{m}} \\ \Theta\left(\frac{1}{\lambda t}\right) & \frac{1}{\lambda\sqrt{m}} \le t \le \frac{1}{\lambda} \\ \Theta(1) & t \ge \frac{1}{\lambda}. \end{cases}$$

Correspondingly, if we approximate the number $m'$ of nonzero coordinates as $\|W_i\|_1^2$, it will start out at $m$, decrease as $1/(\lambda t)^2$, then reach 1 at training time $t = \Theta(1/\lambda)$.

**Numerical simulations**   In Figure 2 we compare our theoretical predictions for $\|W_i\|_1$ and $m'$ (if the constants hidden in $\Theta(\cdot)$ are assumed to be 1) to their numerical values over training when the interference force is removed. The specific values of parameters are $m := 10^5$ and $\lambda := 10^{-5}$, and the initial weights $W_{ik}$ were generated as i.i.d. mean-0 normals with standard deviation $0.9/\sqrt{m}$.

## 2.5   Interference arbiters collisions between features

What happens when you bring the interference force into this picture? In this section, we argue informally that the interference is initially weak if $m \ge n$, and only becomes significant later on in training, in cases where two of the encodings $W_i$ and $W_j$ have a coordinate $k$ such that $W_{ik}$ and $W_{jk}$ are both large and have the same sign—when that's the case, the larger of the two wins out.

**How strong is the interference?**   First, observe that in the interference force on $W_i$

$$-\sum_{j\ne i}\text{ReLU}(W_i \cdot W_j)W_j,$$

each $W_j$ contributes only if the angle it forms with $W_i$ is less than $90°$. So the force will mostly be in the same direction as $W_i$, but opposite. That means that we can get a good grasp on its strength by measuring its component in the direction of $W_i$. We have:

$$\left(\sum_{j\ne i}\text{ReLU}(W_i \cdot W_j)W_j\right) \cdot W_i = \sum_{j\ne i}\text{ReLU}(W_i \cdot W_j)^2.$$

Initially, each encoding is a vector of $m$ i.i.d. normals of mean 0 and standard deviation $\Theta(1/\sqrt{m})$, so the distribution of the inner products $W_i \cdot W_j$ is symmetric around 0 and also has standard deviation $\Theta(1/\sqrt{m})$. This means that $\text{ReLU}(W_i \cdot W_j)^2$ has mean $\Theta(1/m)$, and thus the sum has mean $\Theta(n/m)$. As long as $m \ge n$, this is dominated by the feature benefit force: indeed, the same computation for the feature benefit gives

$$\left(\left(1 - \|W_i\|^2\right)W_i\right) \cdot W_i = \left(1 - \|W_i\|^2\right)\|W_i\|^2 = \Theta(1)$$

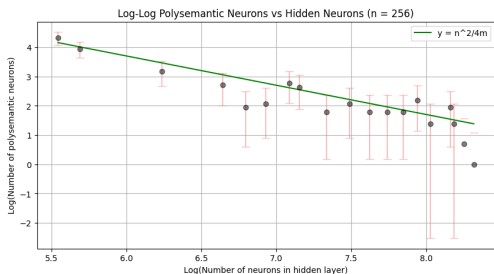

Figure 3: Number of polysemantic neurons against the number of neurons in the hidden layer for 16 different training runs of the non-linear autoencoder with $n = 256$.

as long as $\Omega(1) \leq \|W_i\|^2 \leq 1 - \Omega(1)$. Moreover, over time, the positive inner products $W_i \cdot W_j > 0$ will tend to decrease exponentially. This is because the interference force on $W_i$ includes the term $-\operatorname{ReLU}(W_i \cdot W_j)W_j$ and the interference force on $W_j$ includes the term $-\operatorname{ReLU}(W_i \cdot W_j)W_i$. Together, they affect $W_i \cdot W_j$ as

$$(-\operatorname{ReLU}(W_i \cdot W_j)W_j) \cdot W_j + (-\operatorname{ReLU}(W_i \cdot W_j)W_i) \cdot W_i$$

$$= -(W_i \cdot W_j)\left(\|W_i\|^2 + \|W_j\|^2\right)$$
$$= -\Theta\left(W_i \cdot W_j\right)$$

as long as $\|W_i\|^2, \|W_j\|^2 = \Theta(1)$, which is true at initialization and during training.

**Benign and malign collisions**   On the other hand, the interference between two encodings $W_i$ and $W_j$ starts to matter significantly when it affects one coordinate much more strongly than the others (rather than affecting all coordinates proportionally, like the feature benefit force does). This is the case when $W_i$ and $W_j$ share only one nonzero coordinate: a single $k$ such that $W_{ik}, W_{jk} \neq 0$. Indeed, when that's the case, the interference force $-\operatorname{ReLU}(W_i \cdot W_j)W_j$ - only affects the coordinates of $W_i$ that are nonzero in $j$, - and will probably not be strong enough counter the $l_1$-regularization and revive coordinates of $W_i$ that are currently zero, so only $W_{ik}$ can be affected by this force.

When this happens, there are two cases: (a) If $W_{ik}$ and $W_{jk}$ have opposite signs, we have $W_i \cdot W_j = W_{ik}W_{jk} < 0$, so nothing actually happens, since the ReLU clips this to 0. Let's call this a *benign collision*. (b) If $W_{ik}$ and $W_{jk}$ have the same sign, we have $W_i \cdot W_j = W_{ik}W_{jk} > 0$, and both weights will be under pressure to shrink, with strength $-W_{ik}W_{jk}^2$ and $-W_{ik}^2 W_{jk}$ respectively. Depending on their relative size, one or both of them will quickly drop to 0, thus putting the $k^{\text{th}}$ neuron out of the running in terms of representing the corresponding features. Let's call this a *malign collision*. Polysemanticity will happen when the largest[6] coordinates in encodings $W_i$ and $W_j$ get into a benign collision. This happens with probability $\frac{1}{m} \times \frac{1}{2} = \frac{1}{2m}$, so we should expect roughly $\binom{n}{2} \times \frac{1}{2m} \sim \frac{n^2}{4m}$ polysemantic neurons by the end.

**Experiments:**   Training the model we described on $n \approx 256$ and $m$ ranging from 256 to 4096 shows that this trend of $\Theta\left(\frac{n^2}{m}\right)$ does hold, and the constant $\frac{1}{4}$ seems to be fairly accurate as well. See Figure 3 for more details.

## 3   ANOTHER INCENTIVE FOR SPARSITY: NOISE IN THE HIDDEN LAYER

In the toy model we've considered so far, the encodings were incentivized to be sparse by an explicit $l_1$ regularization term that was added into the loss. While this choice made the toy model very simple

---

[6]This would not necessarily be the largest weight at initialization, since there might be significant collisions with other encodings, but the largest weight at initialization is still the most likely to win the race all things considered.

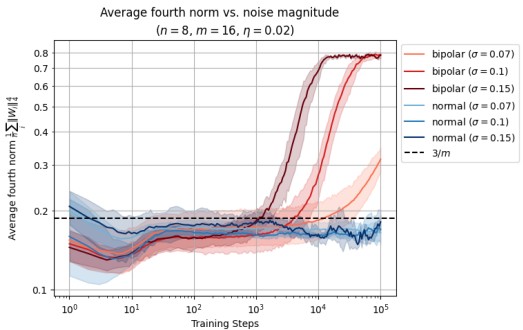

Figure 4: Sparsification process under bipolar and normal noise of various magnitudes. The line $3/m$ is added in as a reference since for large $m$ it is asymptotic to the fourth norm of a random unit vector.

to work with, this is not the most common reason why sparse representations occur in practice. In this section, loosely inspired by Blanc et al. (2020) and Bricken et al. (2023), we show that sparsity can arise when certain types of noise are present in the hidden layer.

### 3.1 MODIFIED MODEL

We will consider a model that's identical to the previous one except that:

- the loss no longer contains the $l_1$ regularization term $\lambda \sum_i \|W_i\|_1$;
- every time the auto-encoder is run, noise from some noise distribution $\mathcal{D}$ is added to each neuron in the hidden layer.

That is, the output is computed as $y := \mathrm{ReLU}\big(W\big(W^\mathsf{T} x + \xi\big)\big)$ for $\xi \in \mathbb{R}^m$, where each coordinate $\xi_j$ is independently drawn from $\mathcal{D}$, and the loss for each input $x$ is defined as

$$\mathcal{L} := \|y - x\|^2 = \left\|\mathrm{ReLU}\big(W\big(W^\mathsf{T} x + \xi\big)\big) - x\right\|^2.$$

Throughout, we will assume that the noise distribution $\mathcal{D}$ is symmetric around 0, has variance $\sigma^2$, and fourth central moment $\mu_4$. Note that this loss is now rotationally symmetric in terms of the hidden layer's space $\mathbb{R}^m$, except for possibly the noise $\xi$: if a rotation were applied before the hidden layer and undone after, nothing would change. In particular, if $\mathcal{D}$ was a normal distribution $\mathcal{N}\big(0, \sigma^2\big)$, the rotational symmetry would be conserved, so there would be no reason for encodings to align with any particular directions. In the remainder of this section, we show through both mathematical analysis and experiments that when the noise $\xi_j$ has negative *excess kurtosis* (which includes many bounded distributions, such as bipolar noise or the uniform distribution over any interval), then encodings will be pushed towards sparsity.

### 3.2 MATHEMATICAL ANALYSIS

In order to make the analysis simpler, we will assume that after $t$ steps of training, the representations are fully learned and there is no interference. More precisely, (a) each encoding $W_i$ has norm $\|W_i\|_2 = 1$; (b) dot product $W_i \cdot W_{i'}$ between pairs of different encodings ($i \neq i'$) is sufficiently negative the noise $\xi$ will not "accidentally turn on" the ReLU's at output coordinate $i'$ when the input is the $i^{\mathrm{th}}$ basis vector: $(W_i + \xi) \cdot W_{i'}$ with high probability.

Concretely, we will compute the update after the $t^{\mathrm{th}}$ step of training, and show that the expected loss at the $(t+1)^{\mathrm{th}}$ step has a term which involves both the fourth norms $\|W_i\|_4$ of the encodings and the excess kurtosis of the noise distribution $\mathcal{D}$.

Since the computations are rather lengthy, we defer the details to Appendix C due to space constraints, but the summary is that:

- Under our hypotheses, we easily obtain that the gradient on input $e_i$ at the $t^{\mathrm{th}}$ step is $\frac{\partial \mathcal{L}}{\partial W_i} = 2(W_i \cdot \xi)(2W_i + \xi)$ (details in Appendix C), and therefore the update is given as $W_i^{(t+1)} = W_i^{(t)} - 2\eta(W_i \cdot \xi)(2W_i + \xi)$.

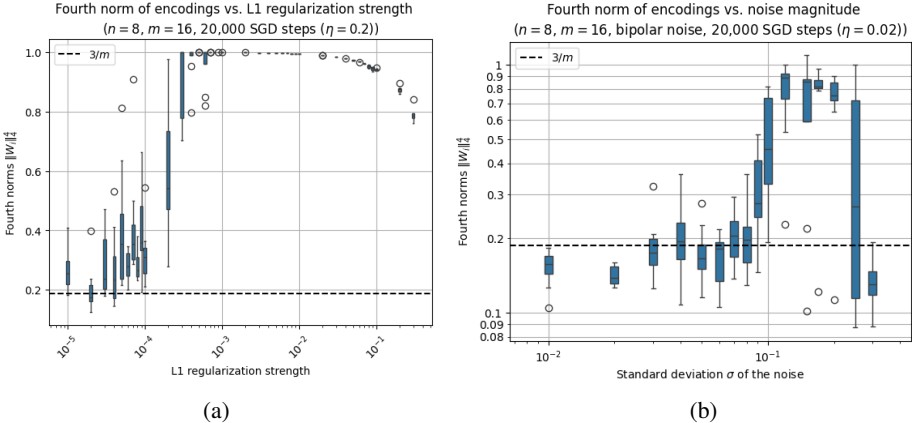

(a)                                (b)

Figure 5: Final fourth norms under $l_1$ regularization and bipolar noises of various magnitudes. The line $3/m$ is added since for large $m$ it is asymptotic to the fourth norm of a random unit vector.

- Plugging this into the error $W_i^{(t+1)} \cdot \left( W_i^{(t+1)} + \xi' \right) - 1$ at the $(t+1)^{\text{th}}$ step, we observe that the expected loss at the $(t+1)^{\text{th}}$ is mostly made out of rotationally symmetric terms (which involve only constants and $l_2$ norms $\|W_i\|_2$) and lower-order terms, but there is one significant and interesting term which appears due to an interaction with the noise at either steps and takes the form $16\eta^2 \mathbb{E}\left[ (W_i \cdot \xi)^4 \right] = 3\sigma^4 16\eta^2 \left( \|W_i\|_2^4 + \|W_i\|_4^4 \left( \mu_4 - 3\sigma^4 \right) \right)$.

Eliminating the rotationally symmetric part, we obtain the implicit regularization-like term $16\eta^2\sigma^4\|W_i\|_4^4 \left( \frac{\mu_4}{\sigma^4} - 3 \right)$, where $\frac{\mu_4}{\sigma^4} - 3$ is the excess kurtosis of the noise distribution $\mathcal{D}$. This means that when $\mathcal{D}$ has negative excess kurtosis, this part of the loss will incentivize $W_i$ to maximize its fourth norm $\|W_i\|_4$, which under the constraint that $\|W_i\|_2 = 1$ means pushing towards sparsity: (a) if $W_{ij} = \pm\frac{1}{\sqrt{m}}$ for all $j$ then $\|W_i\|_4^4 = 1/m$, (b) while if $W_{ij} = \pm 1$ for some $j$ and $0$ elsewhere then $\|W_i\|_4^4 = 1$.

In particular,

- if $\mathcal{D}$ is bipolar noise $\pm\sigma$, which has excess kurtosis $-2$, then this would push towards sparsity;
- if $\mathcal{D}$ is normal noise $\mathcal{N}(0, \sigma^2)$, which has excess kurtosis $0$, then this will not push towards sparsity (and indeed this would maintain the rotational symmetry of the hidden space $\mathbb{R}^m$, and sparsity is not rotationally symmetric).

## 4 DISCUSSION AND FUTURE WORK

Until now, the mechanistic interpretability literature has mostly studied polysemanticity in settings where the encoding space has no privileged basis: the space can be arbitrarily rotated without changing the dynamics, and in particular the corresponding layer doesn't have non-linearities or any regularization other than $l_2$. In such settings, the features can be represented arbitrarily in the encoding space, and we only observe superposition (non-orthogonal encodings) when there are more features than dimensions.

When there is no privileged basis, it is always technically feasible to get rid of superposition by simply increasing the number of neurons so that it matches the number of features. Eliminating polysemanticity that is due to non-task factors could require completely different tools, and seems particularly challenging given that (as we saw in Figure 7), that kind of polysemanticity can happen for a wide variety of sometimes surprisingly hard-to-predict incidental reasons.

In particular, it is much less realistic to do away with the kind of incidental polysemanticity that we demonstrate in Section 2 by simply increasing the number of hidden neurons, since we saw that it

can happen until the number of hidden neurons is roughly equal to the number of features *squared*. On the other hand, since incidental polysemanticity is contingent on the random initializations and the dynamics of training, it could be solved by nudging the trajectory of learning in various ways, without necessarily changing anything about the neural architecture, and this seems like a promising direction for future work.

As a starting point, here is one possible way one might get rid of incidental polysemanticity in a neuron that currently represents two features $i$ and $j$: Duplicate that neuron, divide its outgoing weights by 2 (so that this doesn't affect downstream layers), add a small amount of noise to the incoming weights of each copy, then run gradient descent for a few more steps. One might hope that this will cause the copies to diverge away from each other, with one of the copies eventually taking full ownership of feature $i$ while the other copy takes full ownership of feature $j$.

In addition, it would be interesting to find ways to distinguish incidental polysemanticity from necessary polysemanticity in practice. Can we distinguish them based only on the final, trained state of the model, or do we need to know more about what happened during training? Is "most" of the polysemanticity in real-world neural networks necessary or incidental? How does this depend on the architecture and the data?

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

# Appendix

## A  GENERALITY OF THE MODEL

We chose the toy model in Section 2 to be as simple as possible (and to match Elhage et al. (2022) as closely as possible) while still exhibiting incidental polysemanticity. Nevertheless, in this section, we want to point out that some of these choices are actually without loss of (much) generality.

**Tied weights**  In our model, the encoding and decoding matrices are tied together (i.e. the encoding matrix $W^\top$ is forced to be the transpose of the decoding matrix $W$). This assumption makes sense because even if they were kept independent and initialized to different values, they would naturally acquire similar values over time because of the learning dynamics. Indeed, the $i^{\text{th}}$ column of the encoding matrix and the $i^{\text{th}}$ row of the decoding matrix "reinforce each other" through the feature benefit force until they have an inner product of 1, and as long as they start out small or if there is some weight decay, they would end up almost identical by the end of training.

**Basis vectors as inputs**  If the input features are not the canonical basis vectors but are still orthogonal (and the outputs are still basis vectors), then we could apply a fixed linear transformation to the encoding matrix and recover the same training dynamics. And in general it makes sense to consider orthogonal input features, because when the features themselves are not orthogonal (or at least approximately orthogonal), the question of what polysemanticity even is becomes quite confused.

## B  RIGOROUS ANALYSIS OF THE SPEED OF SPARSIFICATION UNDER $l_1$ REGULARIZATION

For $m' := \#\{k \mid W_{ik} \neq 0\}$, one can write that

$$
\begin{aligned}
-\frac{\mathrm{d}\|W_i\|_1}{\mathrm{d}t} &= \underbrace{\lambda m'}_{\text{regularization}} - \underbrace{\left(1 - \|W_i\|^2\right)\|W_i\|_1}_{\text{feature benefit}} \\
&= \frac{\lambda}{\|W_i\|^2}\left(m'\|W_i\|^2 - \|W_i\|_1^2\right) \qquad\qquad \text{(by balance condition)}\\
&= \frac{\lambda(m')^2}{\|W_i\|^2}\left(\frac{\|W_i\|^2}{m'} - \left(\frac{\|W_i\|_1}{m'}\right)^2\right) \\
&= \frac{\lambda(m')^2}{\|W_i\|^2} \times \underbrace{\frac{\sum_{k:W_{ik}\neq 0}\left(\underbrace{|W_{ik}| - \frac{\|W_i\|_1}{m'}}_{\text{``deviation from mean''}}\right)^2}{m'}}_{\text{``sample variance over nonzero weights''}},
\end{aligned}
$$

where the last inequality is essentially the identity

$$
\mathbb{E}\left[\mathbf{X}^2\right] - \mathbb{E}[\mathbf{X}]^2 = \mathrm{Var}[\mathbf{X}]
$$

where the random variable $\mathbf{X}$ is drawn by picking a $k$ at uniformly at random in $\{\mathrm{co}\{k\} \mid W_{ik} \neq 0\}$ and outputting $|W_{ik}|$.

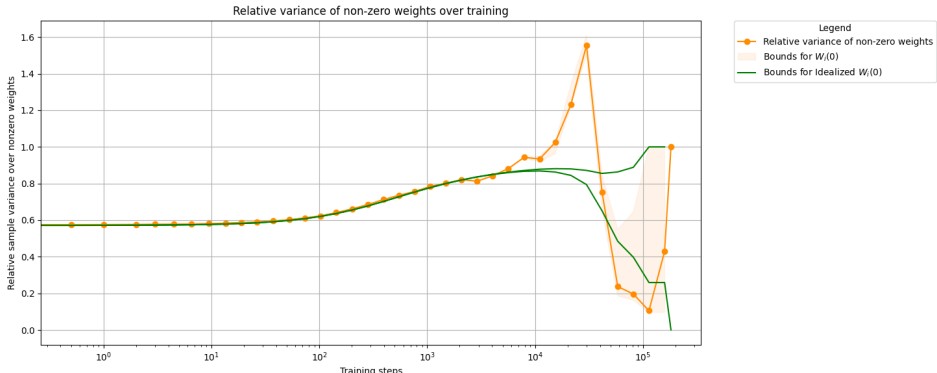

Figure 6: We plot the relative variance over time in the numerical simulation, showing that these lower and upper values for $W_i(0)$ itself (in red) and for an idealized version of $W_i(0)$ that hits regular percentiles (in pink, dashed).

If **X**'s relative variance $\frac{\mathrm{Var}[\boldsymbol{X}]}{\mathbb{E}[\boldsymbol{X}]^2}$ is a constant, then

$$
-\frac{\mathrm{d}\|W_i\|_1}{\mathrm{d}t} = \frac{\lambda(m')^2}{\|W_i\|^2}\mathrm{Var}[\mathbf{X}]
$$
$$
= \frac{\lambda(m')^2}{\|W_i\|^2}\Theta\left(\mathbb{E}[\mathbf{X}]^2\right)
$$
$$
= \Theta\left(\frac{\lambda}{\|W_i\|^2}\|W_i\|_1^2\right)
$$
$$
= \Theta\left(\lambda\|W_i\|_1^2\right), \qquad\qquad \text{(assuming } \|W_i\|^2 = \Theta(1)\text{)}
$$

or if we define $w := \frac{1}{\|W_i\|_1}$ (which is a proxy for the "typical nonzero weight", and is $\approx \theta$ when $\|W_i\|^2 \approx 1$), this becomes

$$
\frac{\mathrm{d}w}{\mathrm{d}t} = \Theta(\lambda),
$$

so $w(t) = w(0) + \Theta(\lambda t)$ and

$$
\|W_i(t)\|_1 = \frac{1}{\Theta\left(w(0) + \lambda t\right)} = \frac{1}{\Theta\left(\frac{1}{\sqrt{m}} + \lambda t\right)}
$$

with high probability in $m$.

Empirically, the relative variance is indeed a constant not too far from 1 (see Figure 6). But why is that?

Suppose that currently $W_{i1} \geq W_{i2} \geq \cdots \geq W_{im} \geq 0$, and let's look at the relative difference between the biggest weight $W_{i1}$ and some other weight $W_{ik} > 0$, i.e.

$$
\gamma_k := \frac{W_{i1} - W_{ik}}{W_{i1}} = 1 - \frac{W_{ik}}{W_{i1}}.
$$

Using logarithmic derivatives, we have

$$
\frac{\mathrm{d}\gamma_k}{\mathrm{d}t} = -\frac{\mathrm{d}(W_{ik}/W_{i1})}{\mathrm{d}t} = -\frac{W_{ik}}{W_{i1}}\left(\frac{\mathrm{d}W_{ik}/\mathrm{d}t}{W_{ik}} - \frac{\mathrm{d}W_{i1}/\mathrm{d}t}{W_{i1}}\right)
$$

Since feature benefit is a relative force, it contributes nothing to the difference of the relative derivatives of $W_{ik}$ and $W_{i1}$, so we just have the contribution from regularization

$$
\begin{aligned}
\frac{\mathrm{d}\gamma_k}{\mathrm{d}t} &= -\frac{W_{ik}}{W_{i1}}\left(\frac{-\lambda}{W_{ik}} - \frac{-\lambda}{W_{i1}}\right) \\
&= \frac{\lambda W_{ik}}{W_{i1}}\left(\frac{1}{W_{ik}} - \frac{1}{W_{i1}}\right) \\
&= \frac{\lambda}{W_{i1}}\left(1 - \frac{W_{ik}}{W_{i1}}\right) \\
&= \frac{\lambda}{W_{i1}}\gamma_k.
\end{aligned}
$$

Note that this differential equation doesn't involve $W_{ik}$ at all! This means that there is a single function $\gamma(t)$ defined by

$$
\begin{cases}
\gamma(0) = 1 \\
\dfrac{\mathrm{d}\gamma}{\mathrm{d}t}(t) = \dfrac{\lambda}{W_{i1}(t)}\gamma(t)
\end{cases}
$$

such that for all $k$, as long as $W_{ik}(t) > 0$,

$$
\begin{aligned}
1 - \frac{W_{ik}(t)}{W_{i1}(t)} &= \gamma(t)\left(1 - \frac{W_{ik}(0)}{W_{i1}(0)}\right) \\
\Rightarrow W_{ik}(t) &= \underbrace{W_{i1}(t)\left(1 - \gamma(t)\right)}_{\text{doesn't depend on } k} \\
&\quad + \underbrace{\frac{\gamma(t)W_{i1}(t)}{W_{i1}(0)}}_{\text{doesn't depend on } k} W_{ik}(0).
\end{aligned}
$$

In other words, the relative spacing of the nonzero weights never change: their change between times $0$ and $t$ is a single affine transformation.

Since the relative variance is scaling-invariant, we can think of this affine transformation as a simple translation. The value of the relative variance of the remaining nonzero weights $W_{i1}(t), \ldots, W_{im'}(t)$ at some point in time must be of the following form:

- take the initial values $W_{i1}(0), \ldots, W_{im}(0)$,
- translate them left by some amount which leaves $m'$ weights positive,
- drop the values that have become $\leq 0$,
- then compute the relative variance of what's left.

In particular, the relative variance when $m'$ weights are left must lie between the relative variance of

$$
\left(W_{i1}(0) - W_{i(m'+1)}(0), \ldots, W_{im'}(0) - W_{i(m'+1)}(0)\right)
$$

and the relative variance of

$$
\left(W_{i1}(0) - W_{im'}(0), W_{i2}(0) - W_{im'}(0), \ldots, 0\right)
$$

(since these extremes have the same variance but the latter has a smaller mean).

These relative variances are functions of $m'$ and the initial value of $W_i$ only, and (when $W_i$ is made of mean-0 normals) they will be $\Theta(1)$ with high probability in $m'$. See the plot (see Figure 6) for a depiction of the lower and upper values for $W_i(0)$ itself (shown in red), and also for an idealized version of $W_i(0)$ that hits regular percentiles (in pink, dashed). The orange curve lies within the red curves, and that the red and pink curves only start to diverge significantly at later time steps when $m'$ is smaller, for reasons detailed above.

## C   GRADIENT AND LOSS COMPUTATIONS UNDER NOISE

### C.1   GRADIENT AT THE PREVIOUS STEP

Let's compute the gradient at the $t^{\text{th}}$ step. To make the math easier to follow, let's temporarily rename the encoding matrix to $W^{\text{e}}$ and the decoding matrix to $W^{\text{d}}$, even though these are the same matrix $W$. For a input $x$, let's consider the values of the hidden layer $h$, the output $y$, the error $\epsilon$ and the loss $\mathcal{L}$:

$$h := (W^{\text{e}})^{\mathsf{T}} x + \xi \qquad\qquad \in \mathbb{R}^m$$
$$y := \text{ReLU}(W^{\text{d}} h) \qquad\qquad \in \mathbb{R}^n$$
$$\epsilon := y - x \qquad\qquad \in \mathbb{R}^n$$
$$\mathcal{L} := \|\epsilon\|^2 \qquad\qquad \in \mathbb{R}.$$

Let $x$ is the $i^{\text{th}}$ basis vector $e_i$. Then

- $h = (W^{\text{e}})^{\mathsf{T}} e_i + \xi = W_i^{\text{e}} + \xi$;

- the output $y$ is 0 everywhere (with ReLUs turned off) except for the $i^{\text{th}}$ coordinate, which is $y_i = W_i^{\text{d}} \cdot W_i^{\text{e}} + W_i^{\text{d}} \cdot \xi = 1 + W_i^{\text{d}} \cdot \xi$, so $\epsilon_i = W_i^{\text{d}} \cdot \xi$;

- $\frac{\partial \mathcal{L}}{\partial o_i} = 2\epsilon_i$ so $\frac{\partial \mathcal{L}}{\partial W_i^{\text{d}}} = \frac{\partial \mathcal{L}}{\partial o_i} \frac{\partial o_i}{\partial W_i^{\text{d}}} = 2\epsilon_i h = 2(W_i^{\text{d}} \cdot \xi)(W_i^{\text{e}} + \xi)$;

- $\frac{\partial \mathcal{L}}{\partial h} = \frac{\partial \mathcal{L}}{\partial o_i} \frac{\partial o_i}{\partial h} = 2(W_i^{\text{d}} \cdot \xi) W_i^{\text{d}}$ so $\frac{\partial \mathcal{L}}{\partial W_i^{\text{e}}} = \frac{\partial \mathcal{L}}{\partial h} \frac{\partial h}{\partial W_i^{\text{e}}} = \frac{\partial \mathcal{L}}{\partial h} I_n = 2(W_i^{\text{d}} \cdot \xi) W_i^{\text{d}}$.

Overall, recalling that $W^{\text{e}} = W^{\text{d}} = W$, we have $\frac{\partial \mathcal{L}}{\partial W_i} = 2(W_i \cdot \xi)(2W_i + \xi)$, and all other gradients are zero on this input. We will see that the part which will push for sparsity is $2(W_i \cdot \xi)\xi$; everything else will either cancel out, almost cancel out, or give rotationally symmetric terms.

By gradient descent, we have $W^{(t+1)} := W^{(t)} - \eta \frac{\partial \mathcal{L}}{\partial W}$, so that for each $i \in [n]$,

$$W_i^{(t+1)} = W_i^{(t)} - 2\eta(W_i \cdot \xi)(2W_i + \xi).$$

### C.2   EXPECTED LOSS AT THE NEXT STEP

At the next step, we get error $W_i^{(t+1)} \cdot \left(W_i^{(t+1)} + \xi'\right) - 1 = \left\|W_i^{(t+1)}\right\|^2 - 1 + W_i^{(t+1)} \cdot \xi'$, where $\xi'$ is the new noise, so the expected loss on input $e_i$ is

$$\mathbb{E}\left[\left(\left\|W_i^{(t+1)}\right\|^2 - 1 + W_i^{(t+1)} \cdot \xi'\right)^2\right]$$

$$= \mathbb{E}\left[\left(\left\|W_i^{(t+1)}\right\|^2 - 1\right)^2\right] + \mathbb{E}\left[\left(W_i^{(t+1)} \cdot \xi'\right)^2\right]$$

$$+ 2\mathbb{E}\left[\left(\left\|W_i^{(t+1)}\right\|^2 - 1\right) W_i^{(t+1)} \cdot \underbrace{\xi'}_{\mathbb{E}[\cdot]=0}\right]$$

$$= \underbrace{\mathbb{E}\left[\left(\left\|W_i^{(t+1)}\right\|^2 - 1\right)^2\right]}_{\text{involves } \xi \text{ only}} + \underbrace{\mathbb{E}\left[\left(W_i^{(t+1)} \cdot \xi'\right)^2\right]}_{\text{involves } \xi \text{ and } \xi'},$$

and we can simplify the second part to

$$\mathbb{E}\left[\left(W_i^{(t+1)} \cdot \xi'\right)^2\right] = \sigma^2 \mathbb{E}\left[\left\|W_i^{(t+1)}\right\|^2\right].$$

Since we've reduced both terms to quantities that involve only $\left\|W_i^{(t+1)}\right\|^2$, let's study it closer:

$$
\begin{aligned}
\left\|W_i^{(t+1)}\right\|^2 &= \|W_i - 2\eta(W_i \cdot \xi)(2W_i + \xi)\|^2 \\
&= \|W_i\|^2 - 4\eta(W_i \cdot \xi)\Big(2\|W_i\|^2 + (W_i \cdot \xi)\Big) \\
&\quad + 4\eta^2(W_i \cdot \xi)^2\Big(4\|W_i\|^2 + 4(W_i \cdot \xi) + \|\xi\|^2\Big) \\
&= 1 - 4\eta(W_i \cdot \xi)(2 + (W_i \cdot \xi)) \\
&\quad + 4\eta^2(W_i \cdot \xi)^2\Big(4 + 4(W_i \cdot \xi) + \|\xi\|^2\Big)
\end{aligned}
$$

First, let's deal with the part which involves the new noise $\xi'$. Because the noise distribution $\mathcal{D}$ is symmetric around 0, we have $\mathbb{E}[(W_i \cdot \xi)] = \mathbb{E}\big[(W_i \cdot \xi)^3\big] = 0$, so

$$
\begin{aligned}
\mathbb{E}&\left[\left\|W_i^{(t+1)}\right\|^2\right] \\
&= 1 - 4\eta(1 - 4\eta)\mathbb{E}\big[(W_i \cdot \xi)^2\big] + 4\eta^2\mathbb{E}\big[(W_i \cdot \xi)^2\|\xi\|^2\big]
\end{aligned}
$$

and $\mathbb{E}\big[(W_i \cdot \xi)^2\big] = \sigma^2\|W_i\|^2 = \sigma^2$, while

$$
\begin{aligned}
\mathbb{E}\big[(W_i \cdot \xi)^2\|\xi\|^2\big] &= \mathbb{E}\left[\Big(\sum W_{ij}\xi_j\Big)^2 \sum \xi_j^2\right] \\
&= \mathbb{E}\left[\Big(\sum W_{ij}^2\xi_j^2\Big) \sum \xi_j^2\right] \\
&= \|W_i\|^2\big(\mu_4 + (m-1)\sigma^4\big)
\end{aligned}
$$

so the part of the expected loss involving both $\xi$ and $\xi'$ is

$$
\sigma^2\big(1 - 4\eta(1 - 4\eta)\sigma^2 \pm 4\eta^2\big(\mu_4 + (m-1)\sigma^4\big)\big),
$$

which is constant and therefore will not push $W_i$ towards or away from sparsity.

Let's now move to the more interesting part, the error that involves only the old noise $\xi$. We have

$$
\left\|W_i^{(t+1)}\right\|^2 - 1 = -4\eta\big(2(W_i \cdot \xi) + \eta(W_i \cdot \xi)^2\big) \pm O(\eta^2),
$$

so

$$
\begin{aligned}
\mathbb{E}&\left[\left(\left\|W_i^{(t+1)}\right\|^2 - 1\right)^2\right] \\
&= 16\eta^2\mathbb{E}\big[4(W_i \cdot \xi)^2 + 4(W_i \cdot \xi)^3 + (W_i \cdot \xi)^4\big] \pm O(\eta^3) \\
&= 16\eta^2\big(4\sigma^2 + \mathbb{E}\big[(W_i \cdot \xi)^4\big]\big) \pm O(\eta^3).
\end{aligned}
$$

The only part which could significantly sway $W_i$ is $16\eta^2\mathbb{E}\big[(W_i \cdot \xi)^4\big]$, and indeed it does:

$$
\begin{aligned}
\mathbb{E}\big[(W_i \cdot \xi)^4\big] &= \sum_j W_{ij}^4\mu_4 + 6\sum_{j \neq j'} W_{ij}^2 W_{ij'}^2\sigma^4 \\
&= \sum_j W_{ij}^4\big(\mu_4 - 3\sigma^4\big) + 3\left(\sigma^2\sum_j W_{ij}^2\right)^2 \\
&= 3\sigma^4\|W_i\|_2^4 + \|W_i\|_4^4\big(\mu_4 - 3\sigma^4\big).
\end{aligned}
$$

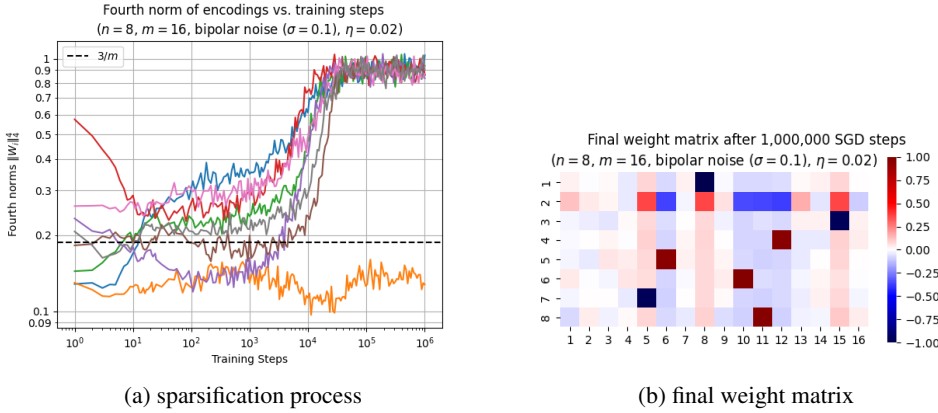

(a) sparsification process          (b) final weight matrix

Figure 7: Sparsification process for a specific instance at $\sigma = 0.01$ of bipolar noise.

## D   COMPARING $l_1$ REGULARIZATION AND NOISE

In this section, we compare the ways that $l_1$ regularization and noise induce sparsity and polysemanticity through various experiments.

In Figure 4 we train autoencoders bipolar and normal noise of various intensities and plot the average fourth norms $\|W_i\|_4^4$ of the encodings as a proxy for how sparse they are. We observe that as expected,

- bipolar noise pushes encodings towards sparsity, and the higher the standard deviation $\sigma$ is, the faster this is;

- on the other hand, in the presence of *normal* noise, there is no observable effect on sparsity, and it only makes the fourth norms oscillate.

In Figure 5, we dig deeper into the effect of the regularization coefficient $\lambda$ (Figure 5a) and the standard deviation $\sigma$ (Figure 5b) on the sparsity after a fixed number of steps. We confirm that regularization and noise of small magnitudes have almost no effect on sparsity and the effect generally grows with magnitude, but the effect from $\sigma$ is much stronger since it appears as a $4^{\text{th}}$ power in the implicit regularization, whereas $l_1$ regularization is linear in $\lambda$. When the regularization and noise get extremely large, we see a drop in the fourth norms due to an overall drop in the magnitudes $\|W_i\|_2$ of the encodings, but the reasons differ slightly:

- when $\lambda$ is very high, the $l_1$ regularization pushes down on all coordinates of each encoding $W_i$ strongly, and once that threshold becomes large enough, the feature benefit force is no longer strong enough to counteract it, even if the encoding $W_i$ is perfectly sparse;

- when $\sigma$ is very high, the direct corruption that the noises incudes on the pre-ReLU output values becomes significant, so the lengths $\|W_i\|_2$ of the encodings are incentivized to shorten.

In Figure 7, we zoom in on a the training dynamics of a typical instance under bipolar noise. In Figure 7a, we separately plot the fourth-norm of each encoding $W_i$, and observe that even though most of the encodings reach almost perfect sparsity (indicated by $\|W_i\|_4^4 \approx 1$), the encoding corresponding to the orange curve seems to be stuck below $\|W_i\|_4^4 = 0.2$. This can be explained by looking at Figure 7b, which visualizes the corresponding final weight matrix $W$. We see that the second encoding row $W_2$ has significant weights in the 7 coordinates that were chosen by the other encodings, and that these weights all have comparable absolute values. What's happening is a fascinating interplay between the interference and the push for sparsity.

- On the one hand, the push for sparsity should incentivize $W_2$ to "pick" one of these 7 coordinates and increase its absolute value at the detriment of the other 6. Indeed, in all cases, the sign of $W_{2j}$ is the opposite of the sign of $W_{ij}$ for the encoding $i$ which maximizes $|W_{ij}|$, so naively, this shouldn't cause any interference.

- But the smaller weights in the matrix $W$ provide a hint to what is actually happening: in each column $j$ for which there is some $i$ with $|W_{ij}| \approx 1$, the other encodings $W_{i'}$ have a small but non-negligible weight with the opposite sign. This is detrimental in terms of the implicit regularization term, but it ensures that the dot product $W_{i'} \cdot W_i$ remains negative (or at least small) even after a small amount of noise is applied to the hidden layer on input $e_{i'}$. If $W_2$ were to choose one of these coordinates $j$, then there would be no such strategy available: indeed, if $W_i$ and $W_2$ were equal the basis vectors $e_j$ and its opposite $-e_j$, then one of $W_{i'} \cdot W_i$ or $W_{i'} \cdot W_2$ must be nonnegative, and changing the value of $W_{i'j}$ in either direction would only make things worse. So $W_2$ is kept from applying this strategy, and is instead forced to compromise between all 7 coordinates in order to keep interference at a minimum.

This is phenomenon is significantly different from the type of polysemanticity that we studied in the previous sections and quite striking, In particular, it explains why the fourth norms were not quite approaching 1 in Figure 4.

