# OpenReview forum: "What Causes Polysemanticity? An Alternative Origin Story of Mixed Selectivity from Incidental Causes"
_ICLR.cc/2024/Workshop/Re-Align — ICLR 2024 Workshop Re-Align Poster_

### Official Review · Reviewer_r4x9 · 2024-02-23
**A nice example of what we can learn with smaller simulations and theoretical analysis**

**Rating:** 2
**Fit:** 3
**Confidence:** 2

**Workshop Review:**

**Summary of Main Contributions**
The authors propose a new hypothesis for the cause of polysemanticity, a phenomenon observed in deep neural networks where multiple unrelated features are represented by the same unit, making mechanistic interpretation of the activity readouts more difficult. The prevailing belief is that polysemantic representations arise out of a necessity to represent the features with a smaller set of neurons, implying it is a problem of capacity. The authors present an alternative explanation that polysemanticity can arise incidentally and demonstrate this through conceptual and mathematical analysis, supported by simulations of idealized models, that at least two other factors (regularization and neural noise) can lead to polysemantic units in networks with more units than inputs. They include an analysis of how many of these incidentally polysemantic units can be expected in a network through an explicit relationship between the number of features and units. Their findings shed new light on the problem of polysemanticity, suggesting it is not just a problem of capacity but perhaps fundamental to the current dominant paradigm of neural networks.

**Strengths**
The authors demonstrate that theoretical analysis and simulation with simple models can yield valuable insights into problems of larger networks.
They not only show the possibility of incidental polysemanticity but that it is guaranteed.
They provide a concrete method to calculate the impact of incidental polysemanticity by an explicit estimate of the number of incidentally polysemantic units that can be expected in a network as a function of the number of features and inputs.
The findings suggest that the problem cannot be solved by capacity increases alone, which is significant.

**Weaknesses**
The mathematical argument is at times hard to follow without decent exposure to the research line. The authors take a somewhat intuitive approach to their analysis, which can lead to confusion if the reader's intuition does not align with the writers'. This makes it hard to assess the validity of the mathematical claims, which rely on a number of nonstandard assumptions and mathematical definitions.
It is not clear whether the authors consider inputs and features as equivalent, and the format of pushing detail to the appendix exacerbates the problem. However, changing the presentation could solve most problems.
The results would be stronger if they were more connected to the broader literature, for example, relating to recent work on sparse autoencoders and interpretability.

**Recommendation**
Accept. The results are intriguing and relevant to the broader community. The methodology, while not perfect, shows that smaller simulations have utility, and the community would benefit from its dissemination.

**Reason For Not Giving Higher Score:**

The logic of the argument is hard to follow and needs some reworking in terms of presentation. This makes assessing the validity of the claims a bit more difficult and would in this stage be more suited for a poster format.

**Reason For Not Giving Lower Score:**

The approach and the results are quite impressive, considering that it relied on theoretical arguments and small models, which is rare to see in the current platying field. The general analysis also gives deeper insight into the nature of polysemanticity.

**Reviewer Domain:**

machine learning

---

### Official Review · Reviewer_wYFr · 2024-02-23
**Interesting problem and interesting results, merits inclusion to the workshop**

**Rating:** 2
**Fit:** 3
**Confidence:** 2

**Workshop Review:**

The paper explores the concept of polysemanticity in deep neural networks and presents a novel perspective on how neurons can represent multiple, unrelated features due to factors like regularization and neural noise, termed "incidental polysemanticity."
This contrasts with the traditional view that polysemanticity arises necessarily from the optimization process in an over-constrained system. The work provides both theoretical and experimental results, which strengthen the claims that polysemanticity can occur even with sufficient neurons to represent features distinctly.

Generally, I think the work is worthy of being included in this workshop.
I would like to note that the references mentioned that propose the traditional origin story for polysemanticity are more contextualized in biological neural networks and real neural data. I am curious how the results of these papers translate to neural architectures in the brain, and are therefore justifiable mechanisms in biology as well.

The implications for AI interpretability and safety are significant, although the authors can do a better job at explaining why mixed selectivity creates an obstacle for interpretability of neural activity. Although I am not an expert in this topic, most experimental data contain a high level of mixed selectivity, yet analysis of that data and its interpretability is not always hindered.

**Reason For Not Giving Higher Score:**

The results are very interesting and well defended, however I do not think their impact and novelty merits a higher score. This might be because they could have better motivated in the paper. I believe that if the authors had implemented one of their future directions mentioned in the discussion section, the one related to imposing "monosemanticity" on a network to make models more interpretable, then the results would have been more impactful. Showing that you can use their origin story to make AI safety related interpretation of models possible would have driven the message home.

**Reason For Not Giving Lower Score:**

The paper is clear, well written, and provides the evidence needed for the claims. The problem and goals are well described, and the authors have a clear contextualization of their work and previous work done. The question tackled is of great relevance to the workshop, and the propose framework is creative, and for lack of a better term "refreshing" as a tackles the old problem of mixed selectivity with a new approach.

**Reviewer Domain:**

neuroscience

---

### Official Review · Reviewer_MkH4 · 2024-02-24
**The work is about the phenomenon of "incidental polysemanticity," and the authors provided theoretical grounds for two causes of the phenomenon.**

**Rating:** 3
**Fit:** 3
**Confidence:** 2

**Workshop Review:**

## Summary
The authors described the phenomenon of "incidental polysemanticity," which is about how neurons in a neural network can accidentally activate for unrelated input features even though the network has enough neurons to avoid reaching the same neuron from two different features. Given a relatively simple autoencoder setup, the authors delivered theoretical grounds for two causes of the phenomenon, L1 regularization and noises in hidden layers, and provided possible solutions showing fair experimental results of numerical simulations.

## Strengths
- The author's argument, based on reasonable assumptions, seemed persuasive through rigorous analysis.

## Weakness
- The one thing I want to point out is the phenomenon's generalizability in a practical setting. All authors' claims were based on a shallow autoencoder model, but the model's architecture seemed very specialized and specific, so it needed to be more generalizable. I understood this is due to analytical convenience. However, I wanted to know how much the author's claim could apply and be generalizable to other models and training schemes.

## Questions and Recommendations
I listed my questions and recommendations in this section, hoping that the following would be constructive and helpful for the authors' future research:
- There always exists a trade-off between interpretability and performance. Can a model's polysemantic tolerance be measured to balance interpretability and performance?
- The author introduced incidental polysemanticity with the assumption that non-task factors might cause it, but how will the phenomena behave differently if the training process contains task factors?

**Reason For Not Giving Higher Score:**

N/A

**Reason For Not Giving Lower Score:**

- The authors' work is strongly connected to the topics of this workshop, and it has a great potential for machine learning theory.

**Reviewer Domain:**

machine learning

---

### Official Review · Reviewer_9BEJ · 2024-02-25
**Presents an interesting look at polysemanticism.**

**Rating:** 2
**Fit:** 3
**Confidence:** 2

**Workshop Review:**

## Clarity
Overall the paper is clear and well-written. There is only one figure which is not particularly clear (font is small for example) and could contain much more information, particularly tying back in to the representations and collisions. But it does clearly show the architecture which is used. The tone of the paper is conversational and this does not help with following the maths. At points coefficients are just dropped or ignored, for example around where it says "Eliminating the rotationally symmetric part...". If this work is going to be submitted for publication then I recommend the authors spend a lot more time on distilling their arguments and formalizing the maths. Space is limited but there are ways for presenting theory work clearly - like with theorems and accompanying proof sketches so that at least we can follow the reasoning. Section 3.2 in particular has a bunch of equations and results just pop out and we can't follow where or how.

## Correctness
Due to some of the clarity issues it is difficult to assess correctness of the maths (I have not checked the appendices). The interpretation of the empirical results seems correct, but in some cases the variance on the results is massive and this is not pointed out or noted fairly. Figure 3 and 5 for example have patches where the predictions are way off and the variance is very large. For example in Figure 5b why does the largest end of the standard deviation result in a huge drop in the fourth norm (contradicting the argument which follows it).

## Novelty and Interest
This work uses an interesting theoretical setup and applies it to an interesting problem. Indeed I think the novelty of the paper is it's greatest strength and the overall discussion it provides on the topic of polysemanticism is clear and interesting. I think this would make it valuable as a workshop piece. The results which are found are also potentially important and likely to spark more work. The fact that sparsity indices polysemanticism is contrary to my intuition and on face value contrary to other findings in the literature. It would be interesting to identify what sets those domains or experiments apart from this work, and the theory presented in this work would certainly help with this.

## Small comments
- "with l1 regularization of parameter λ on the activations" - the maths which follows indicates that L1 is used on the weights as is normal. This like just felt off or misleading.
- The initialization scheme here seems to matter more than it is given credit as. If the norm of the weights are not changing then it indicates that the network is not feature learning (it is in the lazy regime) - this is likely a contributing factor which should be noted.

**Reason For Not Giving Higher Score:**

Clarity and presentation of the mathematics could be improved. Fonts in all figures need to be larger. More descriptive figures or adding more information into Figure 1 could go a long way. Issues with clarity make assessing the correctness or even general intuition difficult.

**Reason For Not Giving Lower Score:**

Problem setup and analysis is novel and interesting. Results are unintuitive and likely to present ample opportunity for discussion and more work.

**Reviewer Domain:**

machine learning

---

### Decision · Program_Chairs · 2024-03-02

Accept (Poster)